# The Role of Extracellular Vesicles in Metabolic Diseases

**DOI:** 10.3390/biomedicines12050992

**Published:** 2024-04-30

**Authors:** Carlos González-Blanco, Sarai Iglesias-Fortes, Ángela Cristina Lockwood, César Figaredo, Daniela Vitulli, Carlos Guillén

**Affiliations:** 1CIBER de Diabetes y Enfermedades Metabólicas Asociadas, Instituto de Salud Carlos III, 28040 Madrid, Spain; carlgo23@ucm.es (C.G.-B.); angelock@ucm.es (Á.C.L.); 2Department of Biochemistry and Molecular Biology, Faculty of Pharmacy, Complutense University of Madrid, Plaza Ramón y Cajal s/n, Ciudad Universitaria, 28040 Madrid, Spain; saraii01@ucm.es (S.I.-F.); cesarfig@ucm.es (C.F.); dvitulli@ucm.es (D.V.); 3IdISSC, 28040 Madrid, Spain; 4Dirección General de Investigación e Innovación Tecnológica (DGIIT), Consejería de Educación y Universidades, Comunidad de Madrid, 28001 Madrid, Spain

**Keywords:** extracellular vesicles, pancreas, liver, pancreatic islets, insulin resistance, cancer, NAFLD, miRNA, diabetic complications, diabetes

## Abstract

Extracellular vesicles represent a group of structures with the capacity to communicate with different cells and organs. This complex network of interactions can regulate multiple physiological processes in the organism. Very importantly, these processes can be altered during the appearance of different diseases including cancer, metabolic diseases, etc. In addition, these extracellular vesicles can transport different cargoes, altering the initiation of the disease, driving the progression, or even accelerating the pathogenesis. Then, we have explored the implication of these structures in different alterations such as pancreatic cancer, and in different metabolic alterations such as diabetes and its complications and non-alcoholic fatty liver disease. Finally, we have explored in more detail the communication between the liver and the pancreas. In summary, extracellular vesicles represent a very efficient system for the communication among different tissues and permit an efficient system as biomarkers of the disease, as well as being involved in the extracellular-vesicle-mediated transport of molecules, serving as a potential therapy for different diseases.

## 1. Introduction

Extracellular vesicles (EVs) are a collection of spheroid structures typically ranging in size from 20 nm to 300 nm, featuring a lipid bilayer membrane that encloses various molecules such as proteins, RNA, or lipids. These vesicles are formed inside a cell called the secretory cell and are released into the extracellular environment, travelling to other target cells where the content of these vesicles is released. This mechanism represents one of the most important types of cell-to-cell communication. These EVs can be classified based on various parameters, such as size, genesis, or content, into three groups: exosomes, microvesicles, and apoptotic bodies (remnants of cells that have undergone an apoptotic process). This review will primarily focus on exosomes and, to a lesser extent, on microvesicles. While the knowledge of these structures dates back to the 1950s, it is only in recent times that the comprehensive exploration of their mechanisms and biological significance has taken place. This is evident in the substantial increase in the number of articles published annually on PubMed with the keyword “EVs” in the title, rising from 31 in 2010 to 2935 by the end of 2023. In addition, it is increasingly common to specify the type of EVs, such as exosomes (mentioned in 281 publications in 2010 to 5096 in 2023) and apoptotic bodies (mentioned in 736 publications in 2010 to 6787 in 2023). New terms are also being created to better specify the function of these EVs, such as oncosomes, which have already accumulated more than 100 publications since this term was created in 2008 to date.

This surge in EV research is primarily attributed to the potential use of these structures as biomarkers for various diseases or as a foundation for the development of more effective treatments compared to existing methods. Furthermore, EVs represent an important strategy for cell–cell communication, transferring information from a donor cell to a recipient cell.

## 2. EVs as Disease Biomarkers

The development of any disease involves a series of changes in various biochemical parameters, such as alterations in the lipid composition of biological membranes, modifications in the production profile of different proteins, or changes in the secretion and signaling of biomolecules. Given that EVs can transport all these types of biological molecules within them, it is logical to consider that the composition of EVs released by secretory cells may also be altered during the development of a disease. As such, they become strong candidates for the discovery of biomarkers for various illnesses.

In this line, various pioneering studies are already analyzing microRNAs (miRNAs), non-coding RNA chains that regulate protein synthesis by blocking the translation of other mRNAs, contained within EVs as biomarkers for neurodegenerative diseases and other diseases. For instance, miR-193b, capable of negatively regulating the expression of amyloid precursor protein, is detected at elevated levels in samples from Alzheimer’s disease patients [1]. Similarly, miR-34a-5p, specifically located in subpopulations of EVs, allows for the differentiation of Parkinson’s disease stages [2]. These studies are not limited to neurodegenerative diseases but also extend to fields such as cardiovascular diseases [3,4], cancer [5], and autoimmune diseases [6].

Additionally, not only are miRNAs being explored as potential disease biomarkers, but also various proteins found within EVs or forming part of their lipid bilayer. An example of this is the analysis of Galectin-3 and Thyroglobulin within EVs in urine, used for prognosis and determining the stage of thyroid cancer. This method is considered potentially more sensitive than the direct serological analysis of these proteins [7]. Likewise, the detection of α-synuclein, tau, and β-amyloid 1–42 in plasma samples from Parkinson’s patients is under investigation [8,9].

### EVs as Biomarkers in Metabolic Diseases

Another of the many utilities that EVs have is to contribute to increasing or reducing the levels of different factors involved in the course of metabolic diseases. It has been shown that adipocyte-derived vesicles (AdEV-derived proteins) can function as mediators in insulin secretion and actively participate in signaling mechanisms related to metabolism [10]. Related to diabetes management, studies have shown that hyperglycemia leads to a decrease in the expression of miRNA-222, which serves as a marker of damage in diabetic retinopathy since it has been proven that. at low levels of this miRNA, there is greater retinal damage and hemorrhage in different layers of the retina [11].

Concerning non-alcoholic fatty liver disease (NAFLD), hepatocyte vesicle-derived miRNAs are more sensitive as biomarkers than direct hepatocyte miRNAs. The decreased vesicle-derived miR-135a-3p expression is related to the course of NAFLD and is, therefore, considered a biomarker for NAFLD [12].

In addition to the previously mentioned techniques for vesicle identification, there is another non-invasive technique implemented in another study based on the use of nano plasmon-enhanced scattering (nPES), which helps analyze biomarkers of the outer membrane of exosomes without the need to isolate them. As a result of this, it has been proven that patients who maintain in plasma high concentrations of small-sized EVs and who have in their outer membrane the asialo-glycoprotein receptor 2 (ASGR2) and cytochrome P450 Family 2 Subfamily E Member (CYP2E1) proteins typical of hepatocytes are related to NASH and the decrease of these proteins and the increase of larger-sized EVs is due to the resolution of the disease [13].

This review aims to analyze and present the most up-to-date information from studies currently underway to (A) understand the physiopathological mechanisms of EVs affecting both the pancreas in the development of cancer and Type 2 diabetes mellitus, as well as the liver in the development of NAFLD; (B) investigate potential modes of pancreas–liver communication through EVs; and (C) examine the therapeutic potential of EVs in these diseases.

## 3. EVs in the Pathophysiology of NAFLD

NAFLD is a metabolic disorder affecting approximately 25% of the global population [14]. It is characterized by the accumulation of fatty acids and triglycerides within hepatocytes, creating a lipotoxic and inflammatory environment in the absence of alcohol consumption. If the presence of fatty acids persists, the severity of the disease increases, leading to a more advanced state known as hepatic steatosis, characterized by hepatocyte ballooning and lobular infiltration. If left untreated, it may progress to cirrhosis and, eventually, hepatocellular carcinoma. The risk of developing the disease increases with factors such as a sedentary lifestyle, hypertriglyceridemia, excessive caloric intake, obesity, and type 2 diabetes mellitus, with the latter appearing to be a central axis in the disease [15]. Insulin resistance gives rise to the “three-hit” hypothesis [16], where triglyceride deposits reorganize in the liver, causing hepatic steatosis. This is accompanied by the accumulation of inflammatory factors, leading to a cyclic process of lipotoxicity, mitochondrial oxidative stress [17], and inflammation. Sustained changes over time contribute to histological alterations, including macrophage infiltration, hepatocyte death, and the progression to advanced and aggressive stages of the disease [18]. Given the involvement of multiple molecules, tissues, and organs in the initiation and progression of the disease, there is a growing body of research proposing the participation of EVs as a potential communication pathway with the liver.

### 3.1. EVs and Adipose Tissue in NAFLD

Adipose tissue, the primary fat depot, acts as an endocrine organ through the secretion of adipokines. White adipose tissue (WAT), particularly linked to lipid storage, undergoes restructuring in situations of overnutrition, leading to an increased number and size of adipocytes. This results in an inability to store more fat, causing ectopic accumulation in various tissues such as the muscle and liver, disrupting glucose metabolism [19]. The increase in adipocyte size is accompanied by the recruitment of macrophages into the tissue (ATMs), classified into two subtypes: M1 with a pro-inflammatory profile (predominant in obese patients) and M2 associated with an anti-inflammatory profile [20]. Recent studies demonstrate the ability of WAT to communicate with the liver, promoting NAFLD progression through exosome-mediated communication. In a study [21], it was concluded, using both approaches in vivo and in vitro, that miRNA-29a secreted by M1 phenotype ATMs obtained from omental adipose tissue of mice induces insulin resistance in hepatocytes via PPARδ, highlighting the involvement of extracellular vesicles in regulating insulin sensitivity. Additionally, Dang et al. [22] proposed an association between insulin resistance and miRNA-141-3p obtained from the adipose tissue of obese mice, noting a decrease in exosome release with this miRNA. This decrease promotes PTEN expression, leading to reduced AKT phosphorylation and decreased glucose absorption in the liver.

In summary, EVs released by white adipose tissue are involved in glucose and lipid metabolism, as well as insulin resistance development through the regulation of specific gene expression or cellular signaling pathways, promoting the development of NAFLD.

### 3.2. EVs and Damaged Hepatocytes

As the liver experiences chronic lipotoxicity and a pro-inflammatory environment, hepatocytes undergo damage, releasing various EVs that exacerbate the disease. In an in vitro study by Zhao, Zhibo et al. [23], EVs loaded with miRNA-122-5p released by a hepatocyte cell line induce the polarization of hepatic macrophages toward an M1 phenotype, leading to the release of pro-inflammatory factors that worsen existing liver inflammation. A similar polarization was observed in an in vivo study [24], where the lipotoxic environment increased miRNA 192-p5, promoting the higher expression of M1 macrophage levels. This polarization appeared to be induced by the negative regulation of Rictor, resulting in the lower expression of p-AKT and p-FoxO. The prolonged presence of the aforementioned factors not only exacerbates the liver’s environment but also promotes the recruitment of components of the immune system, contributing to established inflammation and damage. In an in vitro study [25], Liao CY et al. observed that the liver promoted the migration and adhesion of pro-inflammatory monocytes in hepatic cords in response to vesicles loaded with the sphingolipid S1P. Similarly, Guo, Qianqian et al. obtained similar results, with EVs from obese mice loaded with Integrin β-1 contributing to the chemotaxis and adhesion of monocytes to hepatic sinusoidal endothelial cells [26].

### 3.3. Key Cellular Communication: EVs and NAFLD

Therefore, it can be affirmed that extracellular vesicles play a crucial role in the onset and development of NAFLD. As EVs carry diverse cargo and can be released by different organs to modulate various cellular pathways (Figure 1), understanding their function and identification could be a valuable diagnostic tool for different disease stages. Furthermore, their use as a potential therapy in disease management warrants exploration.

## 4. EV Biomarkers in Hepatocellular Cancer

Techniques such as qT-PCR for the determination of miRNA levels in EVs can be used to characterize different pathologies and their stages. In a study to determine biomarkers of liver cancer, it has been useful to measure miR-224 levels in exosomes. This confirms that patients with advanced tumor stages have a significantly higher concentration of miR-224 compared to controls and that the survival rate decreases the higher the presence of these exosomes in blood serum [27].

In other studies, also using qT-PCR, it has been shown that the expression of miR-320d in exosomes is significantly lower in patients with hepatocarcinoma than in healthy controls. To confirm the presence of these exosomes, TSG101 and CD63 biomarkers, which are found in large quantities in vesicles that carry this miRNA, were quantified in western blot [28]. These results can be useful as the non-invasive diagnosis and prognosis of hepatocarcinoma.

Although the isolation qT-PCR workflow is usually used for the determination of miRNAs in EVs, the technique is unable to differentiate microRNAs that circulate by the secretion from cancerous or non-cancerous cells; therefore, in different types of cancer, attempts have been made to use an alternative approach using a tethered cationic lipoplex nanoparticle (tCLN) biochip to discriminate between EVs coming from tumor cells and those not from tumor cells. It has been shown that exmiR-21 is significantly overexpressed in the plasma of patients with PC compared to the control group, and this technique, unlike others, has the potential to be used as a non-invasive strategy for the early diagnosis of PC [29].

## 5. EVs in Type II Diabetes

Type 2 diabetes mellitus (T2DM), characterized by insulin resistance and declining insulin secretion, affects an estimated 537 million adults worldwide, with a projected increase of approximately 20% by 2030. Early diagnosis and preventive strategies are crucial in order to curb this trend. Recently, the role of EVs in T2DM pathogenesis has gained attention. This review highlights recent advancements in understanding the EV involvement in insulin resistance, and β-cell impairment, and their role in two prominent T2DM complications.

### 5.1. EVs Involved in Insulin Resistance

The pathogenesis of insulin resistance is multifaceted, involving numerous factors. EVs have recently emerged as key mediators of intercellular communication and are increasingly recognized for their role in various pathologies. Numerous studies have investigated the involvement of EVs in insulin resistance, revealing that several tissues, including β cells, adipose tissue, and others, secrete EVs that contribute to the generation of insulin resistance.

β-cells have been shown to secrete EVs containing miR-29, which are taken up by adjacent circulating monocytes. This uptake induces these immune cells to adopt an inflammatory phenotype, thereby promoting insulin resistance through inflammation [30]. Additionally, β-cells secrete mRNA, such as miR-26, which plays a role in regulating insulin secretion and peripheral insulin sensitivity. Reduced levels of miR-26 have been observed in both obese patients and mice. Xu et al. demonstrated that the liver, visceral adipose tissue (VAT), and brown adipose tissue (BAT) can internalize EVs loaded with miR-26, thereby modulating the functions of these peripheral tissues [31].

Guo et al. described a mutation known as Reg1cp found in a long non-coding RNA (lncRNA) of the regenerating genes (Regs). Individuals with this mutation exhibit a higher incidence of T2DM. They found that islet EVs can transfer Mut-Reg1cp into peripheral tissues such as the liver and muscle, triggering insulin resistance by inhibiting adiponectin signaling [32].

Yang Yu et al. reported that miR-27, highly present in the sera of obese patients, induces insulin resistance in skeletal muscle by reducing GLUT4 and insulin receptor expression through the repression of PPARγ [33]. Additionally, other mRNA molecules target PPARγ and contribute to insulin resistance, such as miR-155 and miR-29 secreted by adipose tissue macrophages (ATMs) [21,34]. Furthermore, microRNA-34a, which is highly expressed in the adipose tissue of mice fed a high-fat diet, is transported by EVs into resident macrophages, inhibiting their polarization from a pro-inflammatory to an anti-inflammatory phenotype [20]. These studies demonstrate the crosstalk between adipose tissue and immune cells.

The circulation of microRNAs through EVs is a common mode of intercellular communication. In this context, Castaño et al. observed changes in the plasma mRNA profile of obese mice, which exhibited increased levels of miR-122, miR-192, miR-27a-3p, and miR-27b-3p. The injection of these mRNAs into lean mice induced glucose intolerance and insulin resistance [35].

Apart from microRNAs, there are proteins transported by EVs associated with insulin resistance. Li et al. demonstrated that the knockdown of Sirt1 in the adipose tissue of mice stimulates the secretion of EVs and promotes insulin resistance via the TLR4/NF-κB pathway [36]. Additionally, adipocyte-derived EVs carrying Sonic Hedgehog protein activate macrophages and contribute to insulin resistance [37].

### 5.2. EVs Involved in β-Cell Damage

The maintenance of β-cell mass and function is crucial for regulating insulin levels in the plasma. Any disturbance in these factors can lead to disorders in insulin production and secretion. Studies have implicated EVs as one of the contributing factors to β-cell damage, which can originate from the pancreas or other tissues.

In the context of EVs secreted from the pancreas, research has shown that β-cells exposed to cytokines can modulate their function. For instance, Javeed et al. isolated small EVs from MIN6 cells exposed to a combination of pro-inflammatory cytokines, and then added these vesicles to mouse islets. The exposed islets exhibited β-cell dysfunction through the activation of the CXCL10/CXCR3 axis [38].

On the other hand, EVs originating from cytokine-treated adipocytes, as well as EVs derived from the adipose tissue of obese patients, have been shown to diminish survival and function in rodent and human β cells. These EVs modulate critical signaling pathways in β cells, including PI3K/Akt, ERK1/2, and the unfolded protein response (UPR). Notably, these EVs induce the increased phosphorylation of PERK, eIF2α, and the levels of Chop, promoting a switch in the UPR from an adaptive to an apoptotic role [39].

The intricate crosstalk between β cells and adipose tissue is essential and multifaceted. Ge Q et al. demonstrated that treatment with EVs from obese or insulin-resistant donors decreased β-cell proliferation, increased the phosphorylation of NFkB, and elevated the expression of CCL-2 compared to lean donors [40].

In addition to their role in β-cell damage, EVs also participate in the compensatory mechanism observed in T2DM, which involves an increase in β-cell proliferation in response to insulin resistance. For instance, EVs isolated from the muscle of mice fed a high-palmitate diet were found to stimulate the proliferation of MIN6B1 cells and islets through miR-16 [41]. Moreover, EVs released from adipose tissue macrophages in obese mice can inhibit insulin secretion and increase β-cell proliferation via the involvement of miR-155 [42].

### 5.3. EVs and the Most Common Complications of T2DM

#### 5.3.1. Diabetic Nephropathy

Diabetic nephropathy (DN) stands out as the most prevalent microvascular complication associated with diabetes. It represents a chronic condition characterized by a progressive loss of kidney function attributed to the prolonged exposure to high glucose levels. Individuals affected by DN typically exhibit glomerular hypertrophy, proteinuria, and renal fibrosis.

Recent research has unveiled the significance of inter-tissue communication via EVs in the advancement of DN. For instance, Juan Jin et al. demonstrated that adipose-derived stem cells (ADSCs) release EVs enriched with miR-486, which are internalized by podocytes. This uptake results in the downregulation of Smad1 expression, potentially inhibiting mTORC1 signaling and, thereby, attenuating the progression of DN. These findings suggest that ADSC transplantation holds promise for alleviating renal injury associated with DN [43].

Furthermore, renal cells have been implicated in the modulation of inflammation. Specifically, Lin-Li Lv et al. demonstrated that tubular epithelial cells (TECs) secrete EVs containing miR-19b-3p, which are internalized by macrophages. Subsequently, miR-19b-3p inhibits the expression of SOCS-1, a negative regulator of NF-κB, thereby promoting an inflammatory phenotype in macrophages [44]. This discovery is significant as miR-19b-3p levels may correlate with tubular inflammation.

Additionally, intercellular communication among renal cells has been observed. Xiaoming Wu showed that glomerular endothelial cells (GECs) exposed to high-glucose conditions release EVs enriched with TGF-β1 mRNA, which induces the endothelial–mesenchymal transition (EMT) in podocytes, ultimately leading to barrier dysfunction [45]. Furthermore, TGF-β1 mRNA carried by EVs and secreted by macrophages can activate the proliferation of mesangial cells [46].

Further insights into renal cell communication have been provided by Ravindran S et al., who demonstrated that microparticles released from renal proximal tubular cells (RPTCs) under high-glucose conditions activate the EMT, ERK, and mTORC1 pathways in naive RPTCs [47].

#### 5.3.2. Diabetic Peripheral Neuropathy

Diabetic peripheral neuropathy (DPN) represents a prevalent complication of diabetes characterized by sensory neuropathy with distal motor involvement. During the early stages of DPN and throughout the disease progression, there is an observed upregulation of immune molecules, suggesting a pivotal role of immune modulation in its pathogenesis. EVs have emerged as influential modulators of immune cells, thereby potentially contributing to the progression of DPN.

Despite the lack of effective treatments for DPN, various options, including EV-based therapies, have been proposed. For instance, studies have demonstrated that exosomes derived from mouse mesenchymal stromal cells (MSCs) exhibit the ability to diminish pro-inflammatory cytokines in a mouse model of DNP. These MSC-exosomes are enriched with let-7a, miR-23a, and miR-125b [42].

Furthermore, Schwann cells play a crucial role in synthesizing the myelin sheath of peripheral nervous system axons, and the dysfunction of these cells has been linked to DPN. Notably, the systemic administration of Schwann-cell-derived EVs to mice with DNP has been shown to ameliorate neurological function and reduce sciatic nerve damage [48]. However, it is noteworthy that EVs derived from Schwann cells stimulated with high glucose have been found to suppress the axonal growth of dorsal root ganglia neurons and promote DNP in diabetic mice when administered over 7 days [49].

## 6. EVs in Pancreatic Cancer

Pancreatic cancer can affect both the endocrine and exocrine zone of the pancreas. Tumors that affect the endocrine zone are called neuroendocrine tumors, such as insulinoma, but the majority affect the exocrine part, among which pancreatic ductal adenocarcinoma (PDAC) stands out, both for its incidence and mortality rate, and is the focus of this study. It is a solid tumor formed by different cellular populations, cancer-associated fibroblasts (CAFs), pancreatic cancer cells (PCCs), pancreatic stellate cells (PSCs), and immune system cells such as tumor-associated macrophages (TAMs) whose intercellular communication is key to tumor progression [50,51,52]. It is known hat cancer patients have more circulating EVs than healthy individuals indicating the existence of oncogenic mechanisms that regulate their formation and secretion [53]. EVs have been proposed as one of the mechanisms used by tumor cells to communicate, being the most studied, among microvesicles and exosomes [54]. Uptake can occur through endocytosis or membrane fusion, or mediated by some receptor, and they may contain only signaling molecules and act on the cell surface or release their contents into recipient cells [55]. EVs can carry different molecules involved in the regulation of different processes such as the tumor immune response, angiogenesis, invasive behavior, metastasis processes, and even drug resistance [51]. Most relevant molecules are miRNAs, long non-coding RNAs, and different kinds of metabolites [56].

Various tumor cell types produce EVs that have been shown to suppress natural killer cells, recruit neutrophils, increase STAT3 expression leading to reduced T-cell activation, and induce the apoptosis of CD8^+^ lymphocytes. Specific in vivo studies supporting these findings include those by Han et al., 2022 [57] and McAndrews et al., 2019 [54].

Cancer-cell-derived EVs contain potent proangiogenic factors such as IL-6 and vascular endothelial growth factor (VEGF), along with other molecules capable of enhancing endothelial cell invasion and organization into tubule-like structures. Furthermore, enhanced vascularization promotes tumor dissemination into circulation and, consequently, metastasis, as demonstrated in various in vivo studies [58,59].

Migratory and invasive behavior can be transferred to non-invasive cells via EVs, as demonstrated by various in vivo studies. Molecules enclosed within EVs can dictate tropism for different organs during metastasis, primarily through the expression of integrins, regulatory proteins involved in its uptake [60]. Moreover, some pancreatic-stellate-cell-derived EVs contain miRNAs, including miR-21, that increase cancer proliferation, EMT, and migration [55]. Other molecules involved in tumor cell aggressiveness and proliferation are annexin A6 (ANXA6) and adrenomedullin, which increase proliferation and invasion, both released by pancreatic stellate cells via EVs [56].

Some EVs contain miRNAs involved in drug resistance, such miR-34 and miR-200, with gemcitabine resistance being widely described. Moreover, there are some metabolites such as snail binding protein baculoviral, also carried in EVs, that have been linked to chemoresistance in patients’ tumors [56].

## 7. Connection between Pancreas and Liver

Tissue communication is a complex network of signals that permits the homeostasis of the organism. As has been previously mentioned, EVs represent an interesting strategy to transfer different components from a producing cell to a recipient cell, controlling different parameters. EVs derived from the liver act on different peripheral tissues functioning as endocrine factors, restoring metabolic homeostasis [61]. Although EVs represent a novel intercellular and inter-organ communication capacity in metabolic diseases such as T2DM, in this part of the review, we will specifically focus on the connection between the liver and pancreas. Originally, factors generated by the liver were known as hepatokines, proteins secreted from the hepatocytes for regulating energy homeostasis. Many of these factors have been associated with the appearance of metabolic alterations [62]. In the case of T2DM, which is characterized by an insulin-resistant state and a concomitant increase in pancreatic β-cell mass, it suggests the existence of an inter-organ crosstalk between the tissue in which insulin resistance is originated and the answer in a distant tissue mediated. This link could be mediated by multiple molecules including proteins, microRNAs, certain metabolites, etc. [63]. In this regard, under an insulin-resistant environment, EVs derived from the muscle could induce pancreatic β-cell proliferation through the miRNA-16 [41]. Moreover, several hepatic miRNAs, packaged inside EVs, could improve insulin sensitivity and stimulate pancreatic β-cell proliferation during obesity and insulin resistance, promoting compensatory islet hyperplasia [64,65,66]. Using the transference of the hepatocyte EVs derived from obese mice fed with a high-fat diet into the pancreatic β-cell line MIN6 promoted its proliferation but did not affect the secretion capacity of the cells. A key factor involved in the proliferative capacity of MIN6 was the presence of the miRNA-7218-5p in the liver EVs [64]. In addition, during NAFLD, it is associated with insulin resistance and T2DM; the liver secretes multiple hepatokines and most of them are included into EVs [67]. Among the miRNAs produced by the liver under NAFLD and released in EVs decreasing pancreatic β-cell secretion are the following: miRNA-375 [68], miRNA-9 [69], and miRNA-143 [70]. Very importantly, this interaction operates as well in the opposite direction. Some miRNAs produced in pancreatic β cells and secreted in EVs also exert effects on the liver. For instance, miRNA-26a prevents the metabolic hepatic alterations induced by obesity as well as insulin resistance [31]. In addition, miRNA-29s regulate liver insulin sensitivity and control glucose homeostasis [71]. The EVs obtained from hepatocytes under steatotic conditions promote pancreatic β-cell apoptosis, which is mediated by the miRNA-126a-3p, linking NAFLD and diabetes [72]. The main interactions between the liver and pancreas are resumed in Figure 2.

## 8. Possible Therapeutic Approaches Using EVs

Due to their inherent structure, EVs have also been proposed as novel tools for administering various treatments. This is attributed to their ability to encapsulate different molecules that, due to their natural characteristics, would otherwise be unable to exert their function (either due to their low solubility in biological media, their rapid degradation, or the low capacity of cells to uptake them), transporting them to specific target cells. Additionally, it has been studied that certain types of cells can release exosomes loaded with different types of molecules which, collectively, can have a therapeutic effect. An example of this is the use of exosomes purified from mesenchymal stem cells, containing microRNAs, cytokines, and other factors, showcasing a significant anti-inflammatory capacity. These exosomes can be employed in the treatment of diseases with a high inflammatory component, such as COVID-19 infection or acute lung injury [73,74,75,76,77,78,79].

Some studies have focused on the development of vesicles to inhibit the process of tumor angiogenesis. It has been shown that the inhibition of miR-874 expression is related to the pro-angiogenic effects of tumor cells, and the in vitro and in vivo administration of human liver stream-like cell-derived-EVs. together with this miR-874. significantly inhibits tumor cell angiogenesis, preventing their development [80].

Other studies have focused on decreasing the expression of proteins that collaborate with tumor development. One example is the CD47 protein, which can be found overexpressed in many types of cancer and is related to the malfunctioning of macrophages and dendritic cells and is, therefore, an indicator of poor cancer survival. For this purpose, it has been possible to create vesicles with signal regulatory protein alpha (SIRPα), which, in addition to lowering CD47 levels, increases the secretion of IFN-γ and conditions the infiltration of CD8^+^ and CD4^+^ T cells, further avoiding tumor growth [81].

In diseases that lead to liver failure, attempts are being made to find a possible treatment by isolating exosomes from bone marrow mesenchymal stem cells to protect liver cells from apoptosis, since it has been proven that these exosomes induce the activation of autophagy, releasing autophagosomes and autolysosomes and generating an increase in LC3B II, the anti-apoptotic protein bcl-2, and a decrease in the pro-apoptotic protein caspase-3 (cleaved caspase-3) [82]. The most relevant use of EVs as potential therapeutic agents has been resumed in Table 1, including the mechanism of action involved, as well as the function.

## 9. Challenges of EV Research in Metabolic Diseases

One of the biggest challenges of EV research related to metabolic alterations is the detection of the initial steps of the disease as well as its progression using different biomarkers. These biomarkers could be obtained from different biofluids such as the blood, urine, saliva, etc., and could contain structures such as DNA and RNA molecules, proteins, enzymes, and various types of EVs. For instance, this is important in T2DM and this strategy is called the liquid biopsy and its use is increasingly demanded, not only in T2DM but in other diseases such as cancer and neurodegeneration [83,84]. Very importantly, these EVs must be studied at different levels, including proteomic, metabolomic, and transcriptomic approaches, for a better comprehension of the transported cargoes and how they could modulate the different recipient cells [85,86,87]. Another important challenge is the understanding of the complex heterogeneity of extracellular vesicles that cells can produce with its different biogenesis and function under physiological conditions and different pathologic situations, contributing to a deeper understanding of the disease either in the initiation or during its progression [88].

## 10. Conclusions

In summary, EVs represent an extraordinary mechanism of cell-to-cell communication, playing pivotal roles in both physiological and pathological processes. They are essential for regulating homeostasis by transporting various metabolites, proteins, mRNA, and other molecules. However, depending on the cell type and the cargo they carry, EVs can modulate and contribute to the progression of metabolic diseases such as insulin resistance, beta cell damage in type 2 diabetes, and the accumulation of peripheral free fatty acids in NAFLD. Additionally, in cancer, EVs have been shown to promote cancer cell survival, invasive behavior, angiogenesis, and drug resistance by transporting various metabolites.

However, several aspects remain to be elucidated. One of the most important is determining the specific origin of the EVs detected in different biofluids such as blood/plasma, urine, saliva, etc. Another critical aspect is understanding the differential composition of proteins, lipids, miRNAs, and other molecules within EVs under different physiological and pathological conditions. Lastly, identifying the molecular determinants that precisely target EVs from the producing cell to the recipient cell is crucial for a comprehensive understanding of EV-mediated communication.

## Figures and Tables

**Figure 1 biomedicines-12-00992-f001:**
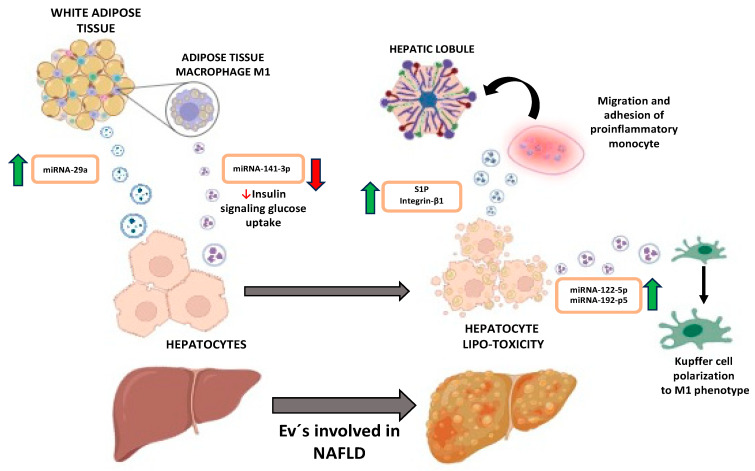
Image depicting the interconnection between adipose tissue and liver. S1P refers to the sphingolipid, sphingosine-1-phosphate.

**Figure 2 biomedicines-12-00992-f002:**
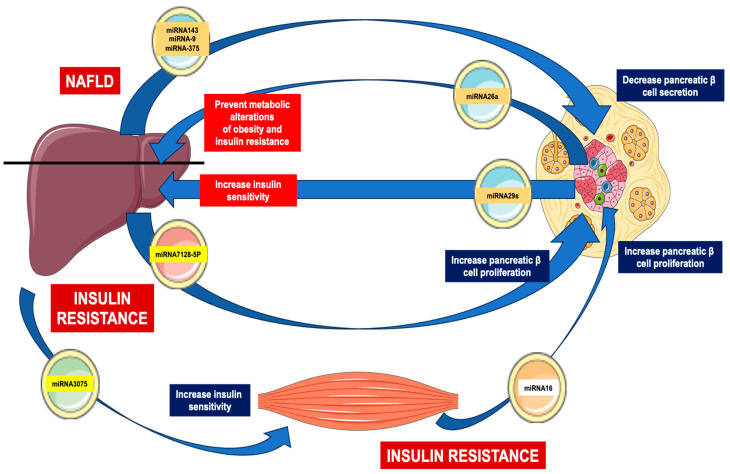
Inter-organ communication between liver and pancreas. NAFLD refers to non-alcoholic fatty liver disease.

**Table 1 biomedicines-12-00992-t001:** Some EVs as possible treatments, their mechanism and function.

EVs	Functions	Mechanism	Reference
Human liver stream-like cell-derived EVs miR-874	Inhibits tumor angiogenesis	STAT3 and VEGF-A suppression	[81]
SIRP alpha	Suppresses tumor growth	Increases IFN-γ and T cell CD8^+^, CD4^+^	[82]
Bone marrow mesenchymal stem cell	Increase autophagy	Increases LC3II, bcl-2 autolysosomes and autophagosomes expression	[83]

## Data Availability

No new data were created or analyzed in this study. Data sharing is not applicable to this article.

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
