# Peer review of "The Role of Extracellular Vesicles in Metabolic Diseases"

_biomedicines, 2024, doi:10.3390/biomedicines12050992_

Round 1

Reviewer 1 Report

Comments and Suggestions for Authors

Overall, this work is an interesting summary on the role of EVs in metabolic diseases. The figures prepared in the manuscript are appealing and catch the attention of the reader. 

Nevertheless, I have a few remarks which need to be addressed prior to publication.

Major:
- The authors should put more emphasis on the evidence of the data: is this data from samples from liquid biopsies or also in vitro data? Human data versus lab animals. I think that should be well discussed throughout the manuscript because the data cannot be easily transferred between species,. 
- The authors should also include a section where they discuss the challenges of EV research especially in metabolic diseases and outlook for future. 
- Authors use often the terminology exosomes throughout the text. Are the referenced papers really about exomes (i.e. purely vesicles derived from the endocytic route), otherwise the terminology "EVs" would be better suited. 

Minor
- Line 40: substantial increase in the amount of publications with EVs as keywork: do the authors also account for other names or terminologies (which might be interesting as well, exosomes, oncosomes, apoptotic bodies)"? 

Comments on the Quality of English Language

Careful proofreading should be performed to improve te quality of the manuscript. A few spelling mistakes are present in the text. Also note to use the correct terminology (EVs instead of exosomes throughout the text according to the latest MISEV guidelines).

Author Response

Overall, this work is an interesting summary on the role of EVs in metabolic diseases. The figures prepared in the manuscript are appealing and catch the attention of the reader. 

Thanks a lot to the referee for her/his kind words about our review. We appreciate a lot these comments. 

Nevertheless, I have a few remarks which need to be addressed prior to publication.

Major:
- The authors should put more emphasis on the evidence of the data: is this data from samples from liquid biopsies or also in vitro data? Human data versus lab animals. I think that should be well discussed throughout the manuscript because the data cannot be easily transferred between species,. 

Thanks a lot for this comment. We have included in the manuscript the evidence of the data, indicating the origin of the samples, which is extremely important for the discussion of the results.  
- The authors should also include a section where they discuss the challenges of EV research especially in metabolic diseases and outlook for future. 

Thanks a lot for this suggestion. We have included in the revised version of the manuscript the new section 8 (Challenges of EV research in metabolic diseases), as suggested by your comment
- Authors use often the terminology exosomes throughout the text. Are the referenced papers really about exomes (i.e. purely vesicles derived from the endocytic route), otherwise the terminology "EVs" would be better suited. 

Thanks a lot for this important comment. We have revised the manuscript and we have corrected all the references to exosomes by the most correct term extracellular vesicles.

Minor
- Line 40: substantial increase in the amount of publications with EVs as keywork: do the authors also account for other names or terminologies (which might be interesting as well, exosomes, oncosomes, apoptotic bodies)"? 

Thanks a lot for the comment. We have included in the revised version of the manuscript the number of publications related to the following terms; exosomes, oncosomes and apoptotic bodies (lines 47-49).

Reviewer 2 Report

Comments and Suggestions for Authors

Comments to the Author and Editor biomedicines-2974842 “The role of extracellular vesicles in metabolic diseases”

During review, the following concerns arose:

1.    The journal's Instructions for Authors indicates: “Acronyms/Abbreviations/Initialisms should be defined the first time they appear in each of three sections: the abstract; the main text; the first figure or table. When defined for the first time, the acronym/abbreviation/initialism should be added in parentheses after the written-out form.”

Authors should check abbreviations throughout the manuscript. It gives the impression that it has been written in parts and then not revised as a whole.

2.    Two figures and a table have been included but are not cited in the text. Define the abbreviations used in each figure in the figure caption.

3.    Page 5, lines 177 to 189. This paragraph that talks about pancreatic cancer, I think, should not go under 4. EVs biomarkers in hepatocellular cancer.

Author Response

During review, the following concerns arose:

  1. The journal's Instructions for Authors indicates: “Acronyms/Abbreviations/Initialisms should be defined the first time they appear in each of three sections: the abstract; the main text; the first figure or table. When defined for the first time, the acronym/abbreviation/initialism should be added in parentheses after the written-out form.”

Authors should check abbreviations throughout the manuscript. It gives the impression that it has been written in parts and then not revised as a whole.

Thanks a lot for the comment and we are sorry about the mistake. We have revised the whole manuscript to detect all the possible alterations to the acronyms defined for the first time

  1. Two figures and a table have been included but are not cited in the text. Define the abbreviations used in each figure in the figure caption.

Thanks a lot to the reviewer for her/his comment. We have included a citation to all the figures and the table included in the manuscript. In addition, we have defined the abbreviations used in each figure.

  1. Page 5, lines 177 to 189. This paragraph that talks about pancreatic cancer, I think, should not go under 4. EVs biomarkers in hepatocellular cancer.

Thanks a lot for this comment and we are sorry about this mistake. Indeed, this paragraph should not be there. Then, we have eliminated this paragraph talking about pancreatic cancer

Round 2

Reviewer 1 Report

Comments and Suggestions for Authors

Thanks you for taking in account the suggestions. I think this work is interesting and represents a nice overview of the state of the art of EVs in metabolic diseases. I would agree with publication in current form. 

Comments on the Quality of English Language

The quality of the English language is good.